

# Combined application of Online FIGAERO-CIMS and Offline LC-Orbitrap MS to Characterize the Chemical Composition of SOA in Smog Chamber Studies

Mao Du[1], Aristeidis Voliotis[1], Yunqi Shao[1], Yu Wang[1], Thomas J. Bannan[1], Kelly L. Pereira[3,†],
Jacqueline F. Hamilton[3], Carl J. Percival[4], M. Rami Alfarra[1,2,‡], Gordon McFiggans[1,*]

[1]Centre for atmospheric science, Department of Earth and Environmental Science, School of Natural Sciences, The University of Manchester, Oxford Road, M13 9PL, Manchester, UK

[2]National Centre for Atmospheric Science, Department of Earth and Environmental Science, School of Natural Sciences, The University of Manchester, Oxford Road, M13 9PL, Manchester, UK

[3]Wolfson Atmospheric Chemistry Laboratories, Department of Chemistry, University of York, York, YO10 5DD, UK

[4]NASA Jet Propulsion Laboratory, California Institute of Technology, 4800 Oak Grove Drive, Pasadena, CA 91109, USA.
[†]Now at: Department of Life and Environmental Sciences, Bournemouth University, Dorest, BH12 5BB, UK.
‡ Now at: Environment & Sustainability Center, Qatar Environment & Energy Research Institute, 34110, Doha, Qatar

[*]*Correspondence to*: Gordon McFiggans (g.mcfiggans@manchester.ac.uk)

**Abstract**. A combination of online and offline mass spectrometric techniques was used to characterize the chemical composition of secondary organic aerosol (SOA) generated from the photooxidation of α-pinene in an atmospheric simulation chamber. The filter inlet for gases and aerosols (FIGAERO) coupled with a high-resolution time-of-flight iodide chemical ionization mass spectrometer (I⁻-ToF-CIMS) was employed to track the evolution of gaseous and particulate components. Extracts of aerosol particles sampled onto a filter at the end of each experiment were analyzed using ultra-performance liquid chromatography ultra-high-resolution tandem mass spectrometry (LC-Orbitrap MS). Each technique was used to investigate the major SOA elemental group contributions in each system. The online CIMS particle-phase measurements show that organic species containing exclusively carbon, hydrogen and oxygen (CHO group) dominate the contribution to the ion signals from the SOA products, broadly consistent with the LC-Orbitrap MS negative mode analysis which was better able to identify the sulphur-containing fraction. An increased abundance of high carbon number (nC≥16) compounds additionally containing nitrogen (CHON group) was detected in the LC-Orbitrap MS positive ionisation mode, indicating a fraction missed by the negative mode and CIMS measurements. Time series of gas-phase and particle-phase oxidation products provided



by online measurements allowed investigation of the gas-phase chemistry of those products by hierarchical clustering analysis to assess the phase partitioning of individual molecular compositions. The particle-phase clustering was used to inform the selection of components for targeted structural analysis of the offline samples. Saturation concentrations derived from

near-simultaneous gaseous and particulate measurements of the same ions by FIGAERO-CIMS were compared with those estimated from the molecular structure based on the LC-Orbitrap MS measurements to interpret the component partitioning behaviour. This paper explores the insight brought to the interpretation of SOA chemical composition by the combined application of online FIGAERO-CIMS and offline LC-Orbitrap MS analytical

techniques.

## 1. Introduction

Secondary organic aerosol (SOA) makes a significant contribution to atmospheric aerosols, which have an important influence on climate and adverse impact on human health and air quality (Nel, 2005; Kroll and Seinfeld, 2008; Hallquist et al., 2009). It is important to

understand the formation, composition and behaviour of SOA due to their contribution to many important atmospheric processes, such as cloud formation (Hallquist et al., 2009). There are more than 10,000 individual organic species in the atmosphere and those almost entirely in the vapour phase. Volatile organic compounds (VOCs) are known to be important in tropospheric ozone and SOA formation (Goldstein and Galbally, 2007). Biogenic volatile organic

compounds (BVOCs) emitted from plants (such as monoterpenes) and anthropogenic volatile organic compounds (AVOCs) (such as aromatics), undergo atmospheric oxidation processes to generate SOA, leading to the formation of a large number of oxidized products (Goldstein and Galbally, 2007; Hoyle et al., 2009; Iinuma et al., 2010; Guenther et al., 2012; McFiggans et al., 2019). As a result of the various VOCs involved and the complexity of the oxidation

processes, oxidation products span a wide range of molecular composition and physiochemical properties (Ma et al., 2008; Laj et al., 2009; Mentel et al., 2015; Mutzel et al., 2015; Mohr et al., 2019). Such chemical complexity poses a major challenge in the molecular characterization of SOA.

Atmospheric simulation chambers play a key role in the study of chemical processes that affect

the composition of the atmosphere, enhancing mechanistic understanding of atmospheric chemistry at a molecular level (Cocker et al., 2001; Rohrer et al., 2005; Wang et al., 2011; Burkholder et al., 2017; Ren et al., 2017). In recent years, new developments in the techniques



and design of laboratory experiments have been carried out to investigate the chemical composition of SOA generated from the biogenic and anthropogenic sources under various

well-controlled conditions, exploring the particle properties, identifying tracers and SOA formation pathways (Winterhalter et al., 2003; Ng et al., 2006; Nguyen et al., 2010; Chhabra et al., 2011; Lopez-Hilfiker et al., 2014; Zhang et al., 2015; Schwantes et al., 2017). Although some progress has been made in characterising sources and formation pathways from some biogenic or anthropogenic precursors, there are still significant gaps remaining in our

understanding of the chemical properties of SOA.

Since there is currently no perfect instrument capable of providing detailed chemical characterisation of SOA (Hallquist et al., 2009; Calvo et al., 2013), various analytical techniques have been established to better characterize SOA chemical composition. Mass spectrometric techniques have been widely employed both online and offline, using a wide

range of sample introduction techniques, ionization methods, and mass analyzers (Hallquist et al., 2009; Nizkorodov et al., 2011; Laskin et al., 2012). Liquid chromatography coupled with electrospray ionization orbitrap mass spectrometry (LC-Orbitrap MS) can probe the chemical composition of individual polar and non-polar particle-phase products with accurate mass measurement (Perry et al., 2008; Banerjee and Mazumdar, 2012; Mutzel et al., 2021). LC-MS

has been extensively used for the chemical characterization of organic aerosols, quantifying targeted tracers and characterising using tandem MS to determine the molecular identity of ambient SOA (Iinuma et al., 2007; Samy and Hays, 2013; Hamilton et al., 2013; Parshintsev et al., 2015; Chen et al., 2020) and laboratory-produced SOA (Winterhalter et al., 2003; Pereira et al., 2014; Mehra et al., 2020b; Mutzel et al., 2021).

Online collection of aerosol samples followed by fast and automated analysis has more recently been developed to overcome some limitations of offline measurements (such as sampling time integration, evaporation, adsorption and potential filter contamination during transportation or storage). One such technique, the Filter Inlet for Gases and AEROsols combined with a soft chemical ionization method and mass spectrometer (FIGAERO-CIMS; Tofwerk A.G. /

Aerodyne Research Inc.), collects filter samples for subsequent thermal desorption to investigate the SOA particle-phase chemical composition (Lopez-Hilfiker et al., 2014). The FIGAERO-CIMS has been coupled to different reagent ions to study the chemical properties of aerosols such as the evolution of gas or particle-phase composition and the volatility of components in laboratory studies and fieldwork (Lopez-Hilfiker et al., 2016b; Stark et al., 2017;


Wang and Hildebrandt Ruiz, 2018; Bannan et al., 2019; Hammes et al., 2019; Lee et al., 2020; Thornton et al., 2020). Other online instruments such as particle into liquid sampler (PILS) coupled with ion chromatography (Bateman et al., 2010) and chemical analysis of aerosol online (CHARON) particle inlet coupled to proton-transfer-reaction time-of-flight mass spectrometry (PTR-TOFMS) instruments (Gkatzelis et al., 2018) can also be employed for

collection and near real-time analysis of chemical compounds in aerosols. Recently, multiple online analytical techniques have been combined to investigate the chemical properties of atmospheric organic carbon, such as the combination of CIMS, PTR-MS and aerosol mass spectrometer (AMS) (Isaacman-VanWertz et al., 2017; Isaacman-VanWertz et al., 2018). There are notable overlaps and differences in the chemical information provided from those

online measurements. Nevertheless, few studies have combined offline LC-Orbitrap MS and online FIGAERO-CIMS to study the molecular composition of SOA (Mehra et al., 2020a). By combining both techniques, the detailed molecular composition can be obtained with time evolution/profiling, offering enhanced mechanistic insights.

The purpose of this paper is to explore the benefits of combinatorial LC-Orbitrap MS and
FIGAERO-CIMS analytical techniques to investigate SOA chemical composition, demonstrating the power of this combination from a technical perspective. To show this experimentally, α-pinene ($C_{10}H_{16}$), as one of the most abundant monoterpenes (emitted in substantial amounts by vegetation, e.g., many coniferous trees, notable pine) was selected to generate SOA in the presence of NOx in the Manchester Aerosol Chamber (MAC). First, both

online and offline analyses were used to retrieve the major elemental group contributions to the SOA in the α-pinene system. Second, temporal profiles of individual components in gas and particle phases provided by online measurement enabled observation of the evolution of identified components. Third, hierarchical clustering analysis (HCA) was employed to reduce the dimensionality of the complex composition measurements by grouping compounds with

similar chemical properties. Finally, saturation concentrations of targeted compounds derived from the partitioning theory in the FIGAERO-CIMS were compared with those derived from the structural information determined using the LC-Orbitrap MS measurements.


## 2. Instruments and methodology

### 2.1 Chamber description and experimental design

Experiments were performed in the Manchester Aerosol Chamber located at the University of Manchester to investigate SOA formation using a broad suite of instrumentation. A full description of the MAC and its characterization can be found in Shao et al. (2021). Briefly, the chamber has a volume of 18 $m^3$ (3m (L) × 2m (W) × 3m (H)) and consists of a fluorinated ethylene Teflon bag which is suspended in the enclosure and supported by three

rectangular aluminum frames. The central frame is fixed, while the upper and lower ones are allowed to move freely to expand and collapse the chamber when sample airflow is introduced to or extracted from the chamber. Relative humidity (RH) and temperature were controlled by the humidifier and air conditioning and were measured at a few points throughout the chamber by a dewpoint hygrometer and a cross-calibrated capacitive sensor. Two 6 kW Xenon arc lamps

and 112 halogen lamps (Solux 50W/4700K, Solux MR16, USA, 16 ×7 rows) are mounted on the wall of the enclosure to provide uniform illumination. The experiments reported here used two arc lamps with quartz glass filters in front of them and 5 rows of halogen lamps to simulate the solar spectra over the wavelength range of 290-800 nm (Alfarra et al., 2012). The reported calculated photolysis rate of $NO_2$ ($j NO_2$), investigated from the steady-state actinometry

experiments, was ~ 0.11-0.18 $min^{-1}$ (Shao et al., 2021).

     The purified dry air was supplied to the chamber by the laboratory air through a dryer (ML180, Munters) and three filters (Purafil (Purafil Inc., USA), charcoal and HEPA (Donaldson Filtration) filters). The chamber was cleaned by a series of automatic fill/flush cycles with purified air (3 $m^3$ $min^{-1}$) before each experiment and by overnight oxidation at a

high concentration of $O_3$ (~1ppm) at end of each experiment. An extra 4-5-hour harsh cleaning experiment under strong ultraviolet photolysis was performed weekly with a high concentration of $O_3$ (~1ppm) to remove the reactive compounds in the chamber. Liquid VOC precursors (α-pinene in this study; Sigma Aldrich, GC grade ≥99.99% purity) are fed into the chamber via injecting into a heated glass bulb (max: 80°C), vaporized immediately and flushed into the

chamber by continuous high purity nitrogen (ECD grade, 99.998 % purity $N_2$). The required concentration of NO$x$ (self-made NOx cylinder, 10% (v/v) $NO_2$ and 90% (v/v) $N_2$) is controlled by the injection of $NO_2$ from a cylinder into the charge line. Seed particles were generated from aqueous solutions using ammonium sulfate (AS, Puratonic, 99.999% purity) by nebulizing via an aerosol generator (Topaz model ATM 230) and introduced into the chamber. During





injections, the seed particles and gases are fed into the chamber with continuous purified air at
flowrate 3 m$^3$ min$^{-1}$, ensuring the rapid and well-mixed in the chamber. The relativity humidity
in the chamber will be adjusted by the custom-built humidifier which comprises a 50L tank fed
with ultra-pure water (resistivity ≥18.2 MΩ-cm), generating water vapour using an immersion
heater that heats the water to ~80°C.

In this work, we will focus on the application of offline and online techniques for the chemical
characterization of gaseous and particulate components formed from the photo-oxidation of α-
pinene in three repeat experiments. To obtain sufficient particulate mass on the filter at end of
each experiment, an initial reference concentration of α-pinene was targeted at 309 ppb for
each experiment, with an initial VOC: NO$x$ ratio ranging from 6 to 8. The α-pinene studies

were part of a series of experiments investigating SOA formation in mixtures and the rationale
behind the experimental design is detailed elsewhere (Voliotis et al., 2021). The O$_3$
concentration was zero at the beginning of each experiment and no additional O$_3$ was added
during experiments. Ammonium sulfate seed aerosol (1g AS dissolving in 100ml deionized
water, ~53±12 μg m$^{-3}$) was added to enable more controllable and uniform condensation of the

gas-phase products and suppress their nucleation. VOCs, NO$x$ and seed particles were injected
into the clean chamber. All experiments were performed under similar conditions (temperature
~ 25°C, relative humidity ~ 50%, 6-h photooxidation reactions). A 1-hour stabilization period
was utilized for measurement of the background conditions in the chamber after injection of
the precursors and seed particles but before illumination of the chamber and the onset of

photooxidation. At the end of each daily experiment, aerosol sample filters were collected
through the 47mm embedded filter holder located at the flushing line of the chamber by pre-
processed filters (Quartz filter: 2.2 μm pore size) at the flowrate of 3m$^3$ min$^{-1}$, before storage
in a freezer at -20°C for subsequent extraction and analysis. Filters were pre-baked at 550°C
for 5.5 h to remove potential contaminants. Table 1 summarises the initial experimental

conditions and instrument availability.

**Table 1.** The initial experimental conditions and instrument availability.

| Exp No | VOC (type) | Seeds | VOC (ppb) | $\frac{VOC}{NOx}$ | Seed concentration (μg m$^{-3}$) | FIGAERO-CIMS availability | LC-Orbitrap MS availability |
|---|---|---|---|---|---|---|---|
| 1 | α-pinene | AS | 309 | 8.4 | *n.a* | × | √ |
| 2 | α-pinene | AS | 309 | 7.7 | 60.7 | √ | √ |
| 3 | α-pinene | AS | 309 | 7.2 | 88.4 | √ | √ |

Note: *n.a* represents no data due to the instrumental issues. The reported initial VOC concentrations in this table have a ± 15% measurement uncertainty.



The repeat experiments were used to evaluate the reproducibility of our experimental system,
our data processing and provide a quantification of our confidence in the processes leading to
the differences in composition. Data differences can arise from a range of sources: control of
experimental conditions, instrumental errors and operational differences, data processing and
error propagation. The treatment of the data from the three experiments from the different
measurement techniques is described in their respective data processing sections.

## 2.2 On-line FIGAERO-CIMS

### 2.2.1 Instrumentation and operation

We employed a FIGAERO-CIMS (Aerodyne Research Inc.) with iodide ($I^-$) as the reagent ion
to measure the gas and particle phase oxidized species produced from photooxidation reactions
of the experiments. This instrument has been described in detail previously (Reyes-Villegas et
al., 2018; Bannan et al., 2019) and was operated at a ~ 4000(Th/Th) resolving power. Gas-
phase species are sampled via a 0.5 m ¼" I.D. PFA tubing at 1 standard litre per minute (SLM)
from the chamber and characterized for 30 mins. Meanwhile, particles are collected
simultaneously in a separate FIGAERO inlet on a polytetrafluoroethylene (PFTE) filter
(Zefluor, 2.0 µm pore size) at the flow rate 1SLM. After 30-min collection, the filter undergoes
thermal desorption (15-min temperature ramp to 200 °C, 10 mins holding time and 8 mins
cooling down to room temperature). Six gas phase and particle phase cycles were carried out
during each experiment. The sample blanks were collected by two additional gas and particle-
phase cycles before the initiation of photooxidation, with the first cycle in the cleaned chamber
condition and followed by the second cycle after all species (VOC, seeds and NOx) injection.
To remove the influence of the seed particles, the particle-phase data collected in the second
cycle were used as the particle-phase background correction. The gas-phase data were collected
in the first cycle was employed for the gas-phase chamber background. The instrument was
flushed with ultra-high purity nitrogen (UHP, 99.999% purity, $N_2$) for 0.2 min every 2 min
during each gas-phase measurement, which acted as the gas-phase instrumental background.

### 2.2.2 Data processing

Data analysis was performed on the Tofware package (version 3.1.2) using the Igor Pro 7.0.8
(WaveMetrics, OR, USA) environment (Stark et al., 2015). Mass spectra were mass calibrated,
after which the high-resolution peaks were fit via the multi-peak fitting algorithm. The exact
mass of the multiple peaks is then matched with the most likely elemental formula, with mass



errors less than ± 6 ppm. Assigned peaks were fitted through the iterative peak assignment method outlined in Stark et. al (Stark et al., 2015). In this study, the *m/z* of the ion adducts with I- range from 200 to 550 was selected to investigate the oxidation products as signals in this range contribute more than 80% of the total signal (exclude the reagent ions, I⁻, $IH_2O^-$, $I_2^-$ and $I_3^-$). All identified molecular formulae were thereafter expressed as $CxHyOzNm$.

Substantial challenges remain in quantifying all identified compounds owing to a lack of standards. The total signal of identified peaks in the range of *m/z* 200-550 (excluded the reagent ions, $I_2^-$: *m/z* 253.809492 and $I_3^-$: *m/z* 380.713964) was used to normalise the ion signal, expressing the relative contribution of each identified compound with an implicit assumption of uniform sensitivity (Isaacman-VanWertz et al., 2018). The gas- and particle-phase dataset

undergoes post-processing in MATLAB (R2017a) code in different ways. For the gas phase, the averaged gas-phase measurement signal of ion *i* was corrected by subtracting the average clean chamber background value and instrumental background value.

For the particle phase, the signal in the first 60 - 90s with relatively low and stable signals was considered as the instrumental background, enabling interference between the gas and particle

mode switching to be removed (Voliotis et al., 2021). All the measured signals were corrected by subtracting the average background value of each ion, after which the temperature of ion counts *i* was integrated to estimate the normalized signal. The thermogram desorption of one compound may have more than one maximum desorption temperature or enhanced tailing even during the temperature ramp period. This can be caused by the fragments from higher

molecular weight compounds or the presence of isomers with different saturation concentrations (Lopez-Hilfiker et al., 2015; Lopez-Hilfiker et al., 2016a; Stark et al., 2017; Lutz et al., 2019). These complex thermogram desorption features for one compound (e.g. two thermogram profiles or enhanced tailing) can be addressed by a custom nonlinear least-squares peak-fitting routine (Lopez-Hilfiker et al., 2015; Lopez-Hilfiker et al., 2016b; Stark et al., 2017).

A non-linear iterative curve fitting method was deployed to fit the peaks (O'Haver, 2021). First, the highest signal was found corresponding to the first maximum desorption temperature (T*max*) of the ion. The peak width was set based on the full width at half maximum (FWHM) and the shape followed by Gaussian distribution. The number of desorption peaks, the location and the amplitude of each peak were optimized to fit the total shape of the desorption profile

and used to retrieve the total ions for each peak. Figure 1 shows an example of desorption peaks of two compounds, which are consistently measured as major components of the detected α-



pinene derived SOA particles. Typically, the results show that the second peak has a T*max* between 20 °C and 90 °C higher than the first peak which is caused by the thermally decomposed accretion products from higher molecular weight rather than isomers (Lopez-
Hilfiker et al., 2015). Thus, the first desorption peak was considered as the monomer generated from the precursor oxidation, and its related integrated ion counts were employed for normalized signal and partitioning calculations.

Since a common peak list is used for two experiments, the normalised fractional CIMS signal for the repeat experiments can be reported as averages along with the associated variability (as
in Figures 2 and 4) or as values from one representative experiment without variability (e.g., Figures 3, 5 and 6) associated with supplementary figures representing another experiment as appropriate.

### 2.2.3 Hierarchical clustering analysis

Hierarchical clustering analysis (HCA) is an analytical method that is used to group or cluster
datasets with a large number of individual observations based on the similarity of their behaviour over time. It is used here to investigate the time series of gaseous and particulate oxidation products measured by CIMS and FIGAERO-CIMS. Generally, there are two main hierarchical clustering analysis methods: divisive hierarchical clustering (working in a top-down method) which describes if a cluster needs to be split and agglomerative hierarchical
clustering (working from the bottom-up) which describes if clusters should be combined (Bar-Joseph et al., 2001; Müllner, 2011; Nielsen, 2016). Compared to divisive hierarchical clustering, agglomerative hierarchical clustering is commonly used to cluster measurements with distinct time-series behaviours from mass spectrometry datasets and describe the degree of similarity between any two measurements, reducing the dimensionality of mass spectrometry datasets
and improving understanding of bulk properties of the chemical processes (Sánchez-López et al., 2014; Rosati et al., 2019; Koss et al., 2020; Priestley et al., 2021). HCA is independent of calibration or instrumental sensitivity since it relies on the relative differences between time series shapes. The final number of clusters is decided based on the distance between the objects and is decided by the user. In this study, we illustrate how HCA can be implemented to reduce
the complexity of a dataset while retaining the chemical information by investigating the oxidation processes or product properties from the photooxidation of α-pinene.



First, to observe the time-series trend changes of each ion and remove the effect of the differences caused by the absolute signal intensity, all measurements are normalized to the highest signal in each time series of each ion. Second, the Euclidean distance between each

pair of points of ion A and ion B was calculated (Eq. 1). The sum of the distance for each pair of observations was considered as the distance metric since this approach is most reproducible and least sensitive to outlier points in the time series (Koss et al., 2020).

$$d_{A,B} = \sum_t abs(A_t - B_t) \qquad\qquad\qquad \text{Eq 1}$$

Third, the average linkage criterion was selected to determine the distance between sets of

measurements as this criterion gave more similar or understandable results. The algorithm starts with the distances between all observations. As a first step, the lowest distance between observations A and B is identified and is assigned a new cluster $x$. The two observations of A and B are removed from the distance dataset and the new cluster $x$ is added into the dataset. Then the distance set between the new cluster $x$ and the remaining observations (named $y$) was

calculated as the average of the distances between each of $n$ individual members of $x$ and $m$ individual members of $y$ over all points $i$ in cluster $x$ and points $j$ in cluster $y$, as shown in Eq. 2. The smallest distance between those observations is then found and made up into a new cluster iteratively until only one cluster remains. Finally, the dendrogram tree is deployed to display the relationships between each measurement and cluster. The whole process is carried

out in MATLAB (MathWorks, Inc.).

$$d_{x,y} = \sum_{i,j} \frac{d_{(x_i, \; y_j)}}{m \times n} \qquad\qquad\qquad \text{Eq. 2}$$

**2.3 Off-line LC-Orbitrap MS**

**2.3.1 Sample preparation**

The sample preparation method described in Bryant et al., (2020) was modified for our

experimental conditions. Briefly, 4 ml methanol was added to a cleaned and dried extraction vial containing half a filter cut into small pieces and left at room temperature for 2h before sonicating for 30 mins. The extractant was then filtered through a 0.22 µm filter (Thermo Fisher Scientific) using a syringe (1mL, BD PLASTIC PAK, STERILE) into another sample vial, which was evaporated to dryness using a vacuum solvent evaporator (Biotage, Sweden). The

sample was reconstituted in 1ml of 10: 90 methanol: water (optima LC-MS grade, Thermo Fisher Scientific) for analysis.



### 2.3.2 LC- Orbitrap MS analysis

The samples were analyzed using ultra-performance liquid chromatography ultra-high-resolution mass spectrometry (Dionex 3000, Orbitrap QExactive, Thermo Fisher Scientific).
Compound separation was achieved using a reverse-phase $C_{18}$ column (aQ Accucore, Thermo Fisher Scientific) with the following dimensions: 100 mm (length) × 2.1 mm (width) and 2.6 µm particle size. The column was held at 40°C during analysis and the samples were stored in the autosampler at 4°C. Gradient elution was used, starting at 90 % (A) with a 1-minute post-injection hold, decreasing to 10 % (A) at 26 minutes, returning to the starting mobile phase
conditions at 28 minutes, with a 2-minute hold to re-equilibrate the column, and the total running time was 30 minutes for each sample. The flow rate was set to 0.3 ml/min with a sample injection volume of 2 µl. Heated electrospray ionization was used, with the following parameters: capillary and auxiliary gas temperature of 320 °C, sheath gas flow rate of 70 (arb.) and auxiliary gas flow rate of 3 (arb.). The mass spectrometer was operated in negative and
positive ionization mode with a scan range of m/z 85 to 750 and mass resolving power of 70000 (Th/Th). Tandem mass spectrometry was performed using higher-energy collision dissociation with a normalized collision energy of 65, 115.

### 2.3.3 Data processing

Data were analyzed by two complementary approaches. The conventional approach for LC-
MS analysis entailed extracting fragments and structural information of targeted compounds using the XCalibur software 4.3. A more recent automated non-targeted approach extracts all detected chromatographic peaks from the sample data set, removing any compounds which do not satisfy a set criterion (see below). The molecular formula is automatically assigned for each compound via a bespoke method designed in the analysis platform, Compound Discoverer
version 2.1 (Thermo Fisher Scientific). Full details of the methodology can be found (Pereira et al., 2021). Briefly, molecular formulae assignments were allowed unlimited C, H, O atoms, up to 2 S atoms and 5 N atoms, plus > 2 Na atoms and 1 K atom in positive ionization mode. Compounds with a mass error < 3 ppm and signal-to-noise ratio > 3, a hydrogen-to-carbon ratio of 0.5 to 3 and an oxygen-to-carbon ratio of 0.05 to 2 were included in the data set. Any
compounds detected in the procedural (control sample, *i.e.* blank pre-conditioned filter subjected to the same extraction procedure) and solvent blanks (instrumental blanks) with the same molecular formula and a retention time difference within 0.1 min were removed from the sample data.  Furthermore, any compounds detected in the chamber background filter with the

same molecular formula and a retention time difference within 0.1 min and the ratio of signal
intensities between the sample data and chamber background lower than 3 were removed from
the sample data. As with online FIGAERO-CIMS, since quantification for all identified
compounds using standards is not possible, all compounds were normalized to the total signal
to express the relative contributions, implicitly ignoring differential sensitivity in comparison
between the two instruments.

The peak list for the LC-Orbitrap MS is not identical for all experiments since some peaks were
not detected in all experiments. The uncertainty as to whether such peaks are truly
representative of particle-phase oxidation products is captured by classifying the peaks as
"common" (common to all repeat experiments) and "inconclusive" (in one specific experiment
or common to two of all experiments) and then by expressing their contribution on average
across the experiments with the uncertainty across each in Figure 2. In some cases, the results
from one representative experiment are used, such as in Figure 3.

**2.4 Elemental grouping of identified compounds**

The identified species from FIGAERO-CIMS mass spectra and from the automated method for
LC-Orbitrap MS analysis described above were grouped according to their elements.
Compounds containing only carbon, hydrogen, oxygen and nitrogen were categorized as the
CHON subgroup. Components containing only carbon, hydrogen and oxygen were classed as
CHO subgroup. Similarly, compounds containing carbon, hydrogen, oxygen and sulfur or
containing carbon, hydrogen, oxygen, nitrogen and sulfur fell into CHOS or CHONS
subgroups respectively. Compounds were additionally classed according to the numbers of
carbon atoms in their molecular formulae: $C_2$ to $C_7$, $C_8$ to $C_{10}$, $C_{11}$ to $C_{15}$, $\geqslant C_{16}$. Given the
selectivity and sensitivity of the ionization methods, not all ions were equally observed in the
two instruments, and thus only the dominant contribution of identified compounds included in
the CHO and CHON groups are discussed in this paper. The minor contribution of CHOS and
CHONS groups are not attributed in the FIGAERO-CIMS measurement as it is difficult to
identify the two species reliably within the trusted error ($\pm$ 6 ppm). Previous studies also
showed difficulties in the identification of isoprene-derived organosulfate compounds from
CIMS measurements unambiguously due to the low mass resolution or thermal desorption of
organosulfate compounds (Xu et al., 2016; D'Ambro et al., 2019). Compounds in the two
groups may be included in the unassigned category or may misattribute their signals to the
CHO and CHON for the online measurement. The unassigned fractions in the FIGAERO-



CIMS measurements (less than 20% of total signal) represent the peaks that are difficult to identify, caused by either the poor signal-to-noise ratios (S/N⩽2, ~40-50%) or inaccessible formulae within the trusted error.

## 2.5 Calculation of average carbon oxidation state ($\overline{OS_C}$)

The average carbon oxidation state is commonly deployed to describe the degree of oxidation within a complex oxidation reaction (Kroll et al., 2011). The elemental ratios of oxygen-to-carbon (O:C), hydrogen-to-carbon (H: C) and nitrogen-to-carbon (N: C) are used to calculate the $\overline{OS_C}$. For the CHON compounds, we assume nitrogen is in the form of nitrate where the oxidation state of N ($OS_N$) is +5 if the oxygen number of the molecule is equal to 3 or more

than 3. The nitrogen is in the form of nitrite where the $OS_N$ is +3 if the oxygen number of the molecule is less than 3. In the FIGAERO-CIMS, the $\overline{OS_C}$ for the CHOS and CHONS compounds is not considered because of the associated challenges in quantifying S-containing compounds (see above). In the LC-Orbitrap MS analysis, the total signal fractions of CHONS and CHOS groups make low contributions to the total signal, accounting for ~10 ± 2 % in the

common peaklist in both modes and for 12.4% ± 2.7% and 28.5% ± 2.5% in the inconclusive peaklist for negative mode and positive mode, respectively. It is also impossible to assess if the structure of a compound in the CHONS should be assigned as nitrooxy-OSs with both –OSO₃H and –ONO₂ groups or as other combinations (e.g. -NO₂ and -SO₃H, -NO₂ and -OSO₃H etc). Given the two considerations, the calculation of $\overline{OS_C}$ for the CHOS and CHONS compounds

is simplified to the equation of $\overline{OS_C} \approx 2\times O/C - H/C$ here. The uncertainty of this assumption on the calculated $\overline{OS_C}$ for CHOS and CHONS species is further discussed in Supplementary Information. The modification of the $\overline{OS_C}$ is shown in Eq. 3.

$$\overline{OS_C} \approx 2\times O/C - H/C - (OS_N \times N: C) \qquad\qquad \text{Eq. 3}$$

Where $OS_N = 0$ if $n$N =0; This is for CHO compounds, and CHOS and CHONS compounds

for LC-Orbitrap MS analysis; $OS_N = +3$ if $n$O <3; $OS_N = +5$ if $n$O >= 3.

## 2.6 Estimation of component properties

Data from both techniques can be used to investigate the gas-to-particle partitioning behaviour of the targeted compounds in the α-pinene precursor system. Saturation concentration ($C^*_{i,Fp}$), the concentration at which 50% of a component, $i$, is in the vapour phase and 50% is condensed,





can be estimated based on the partitioning method from particle-phase fractions ($F_p$) from FIGAERO-CIMS. Although the homologous series of polyethylene glycols (PEG; ($H-(O-CH_2-CH_2)n-OH$) for n=3 to n=8) were performed to calibrate the vapour pressures in the FIGAERO-CIMS, useful polyethylene glycols (PEG; ($H-(O-CH_2-CH_2)n-OH$) for n=3 to n=8) calibrations are not accessible for interpretation of this dataset (see Voliotis et al. (2021)

for a full discussion). The thermogram method for the online measurements is therefore not accessible to this study and the partitioning method is applied here. Additionally, saturation concentration of a component, $i$, ($C_i^*$) was estimated based on the structural information independently obtained from LC-Orbitrap MS.

**2.6.1 Partitioning Method**

$F_p$ is calculated for a given species $i$ by equation 4, from P and G; the total signals are corrected by the sampling volume in the particle and gas phase, respectively. Some instrumental factors can affect the gas to particle partitioning process and thus affect the $F_p$ and C* calculated in this way, such as the deposited mass of the component on the filter (Thornton et al., 2020), gas-particle phase equilibrium on the filter, reactions on the surface or in the bulk, or diffusion of

the gas to the particle-phase surface or the bulk (Mai et al., 2015; Yli-Juuti et al., 2017; Huang et al., 2018).

$$F_{pi} = \frac{Pi}{Pi+Gi}$$    Eq 4

C* can be estimated by applying partitioning theory using the equation

$$C_{i,Fp}^* = OA \times (\frac{1}{F_P} - 1) = OA \times \frac{G}{P}$$    Eq 5

Where OA is the suspended particle concentration in the chamber, which is calculated based on the AMS measurement, $\mu g\ m^{-3}$.

**2.6.2 Molecular structural method**

The structure of targeted compounds is identified using the LC- Orbitrap MS, after which vapor pressure is estimated using the EVAPORATION method (Compernolle et al., 2011) in the

UManSysProp tool developed by Topping et al. (Topping et al., 2016). The EVAPORATION method has been assessed as more accurate than other vapor pressure estimation methods for vapour pressure of individual organic compounds (O'Meara et al., 2014), though significant



uncertainties persist, particularly for multifunctional compounds. The C* is calculated using equation 4 modified by Donahue et. al (Donahue et al., 2006) based on the earlier absorptive

partitioning theory (Pankow, 1994). Errors in the estimation of C* attributable to the uncertainties of vapor pressure estimation are extensively discussed in the literature (Barley and McFiggans, 2010; Bilde et al., 2015), and are not further discussed here. Additional uncertainties related to the activity coefficient in mixtures of components – implicit in the derivation of C* from CIMS and explicitly assumed as unity for the LC-Orbitrap MS.

$$C_i^* = \frac{M_i 10^6 P_{L,i}^\circ}{RT}$$    Eq 6

Where $M_i$: molecular weight of species $i$, g/mol;

$P_{L,i}^\circ$: saturation vapor pressure of pure compound $i$ at temperature T, Pa; here T is 298.15K;

R: gas constant, 8.314 $m^3$ Pa $K^{-1}$ $mol^{-1}$.

## 3. Results and discussion

We used the data processing method described in Sections 2.2 and 2.3 to generate the full formula list from FIGAERO-CIMS and LC-Orbitrap MS measurements, respectively. Assigned chromatographic/mass spectral peaks are used to explore the molecular formulae and broad chemical groupings in the α-pinene photooxidation system. Common capabilities of the two techniques will be illustrated in section 3.1. Additional capabilities of each technique and

the strength of their combination are shown in sections 3.2, 3.3 and 3.4, respectively.

### 3.1 Overview of SOA elemental composition

Offline filters characterized by LC-Orbitrap MS are shown in Fig 2 from negative and positive ionization mode alongside the final gas and particle-phase measurements by the CIMS, 5.5 hours into the photochemistry experiment. In the LC-Orbitrap MS, the common compounds in

all experiments account for ~74% ± 2.5% in the negative mode and ~ 59% ± 8.5% in the positive mode, as shown in Figure 2.

General observations can be drawn from the CIMS analysis, including i) there are smaller molecules and fewer larger ones in the gas phase, ii) compounds in the CHO group dominate the particle phase, and iii) compounds with between 8 and 10 carbon atoms dominate both gas

and particle phases.



From the CIMS particle-phase and the LC-Orbitrap MS common compounds data, the results show that i) the majority of particle-phase signals come from the compounds with 8 to 10 carbon numbers in the CHO group; ii) a small proportion (4% ± 0.8%) of high carbon number compounds ($C_{11}$-$C_{20}$) is observed in the particle phase; iii) the CHO group is shown to account for around 76% ± 2.7% of the total contribution in the CIMS particle-phase signal and ~55% ±1.4% of the total contribution in the LC-Orbitrap MS negative mode; iv) the CIMS particle-phase fractional signal contribution shows high similarity for the $C_8$-$C_{10}$ and $C_{11}$-$C_{15}$ range to the LC-Orbitrap MS negative mode; v) LC-Orbitrap MS negative and positive mode measure a higher fractional signal from compounds with carbon number greater than 16 than the CIMS particle phase, most likely as a result of the higher mass resolution; vi) in contrast, more small molecules with low carbon numbers ($C_2$-$C_7$) are observed from the CIMS particle-phase measurement, which may be due to the thermal decomposition of low-volatility molecules into smaller fragments (Lopez-Hilfiker et al., 2015; Isaacman-VanWertz et al., 2016; Thornton et al., 2020). The fractional particle-phase contributions to the CHO and CHON groupings were comparable between the techniques, though a higher fraction of larger molecules was observed particularly in the CHON fraction in the LC-MS negative mode measurement.

From the LC-Orbitrap MS common compounds alone, the results suggest that the positive ionisation mode shows that the CHO and CHON groups contribute to around 48%±7% of total sample abundance. A large fraction (~11%±2.6%) of compounds in the CHON group was observed to contain a high carbon number ($\geqslant C_{16}$), showing more formulae with high molecular weight in the positive mode. Signals in the CHO group in both the positive and negative modes are dominated by the $C_{8-10}$ ions.

The inconclusive compounds in all experiments from the LC-Orbitrap MS are distributed in the four groups (grey bar in Figure 2a), accounting for 26.1% ± 6.6% and 40.6% ± 6.1% in both modes. As shown in Figure 2b, 56% ± 4.8% of the inconclusive compounds are with high carbon number ($\geqslant C_{16}$) in the negative mode, and the compounds with nC = 8-10 and nC $\geqslant$ $C_{16}$ make large fractions in the positive mode, accounting for ~ 43% ± 4.4% and 33% ± 3.4% to the total inconclusive compounds, respectively. The results suggest that the largest fractions of compounds with high carbon numbers are found in the inconclusive compounds, although the existence of those compounds may result from some uncertain factors, such as control of experimental conditions, instrumental errors and operational differences or data processing. Irrespective of the consideration of the inconclusive compounds, a higher fraction of the signal


is found in compounds with high carbon numbers in the LC-Orbitrap MS measurements than in the CIMS particle measurements.

Measurements of the oxidation products from both techniques in one representative experiment can be represented in terms of three different chemical spaces as shown in Figure 3. Figure 3a shows that more compounds with higher carbon numbers (>10) are observed in the CIMS particle phase compared to that in the gas phase. In the particle phase, Figures 3a and 3b show that the results from LC-Orbitrap MS in both ionisation modes show more molecules with

higher carbon numbers (C≥16) and higher oxygen number molecules (O≥15) than the results from CIMS particle measurement, though with some of them are from inconclusive compounds. More compounds with lower carbon numbers (<5) are mostly detected by the CIMS particle measurements; likely owing to high molecules thermally desorption into small molecules in the online particle-phase measurements or the inability to measure high volatility compounds

using LC-Orbitrap MS.

To explore the chemical properties of those compounds and avoid the overlapped formulae with the same carbon number and oxygen number but different hydrogen number, such as the $C_{10}H_xO_4$, carbon number vs $\overline{OS_C}$ and O:C vs H:C are shown in Figures 3c - 3f. Figure 3c and 3d display that some molecules with low carbon number (C<5) and high $\overline{OS_C}$ ($\overline{OS_C}$ >0) are

detected by the CIMS, while molecules with 10<nC<20 and $\overline{OS_C}$ >0 and molecules with high carbon number (C>20) are only measured by the LC-MS techniques, although some of them are inconclusive compounds. Figure 3e and 3f show that the results from the two instruments are dominated by formulae with 1.3 < H/C < 1.8 and 0.3 < O/C < 0.7 which show in the center of the diagram with high density. The majority of compounds in the left bottom of the diagram

with a low H: C ratio (<1) and O: C ratio (<0.3) are compounds in the S-containing subgroups, which are only identified by LC-Orbitrap MS. Some highly oxidized (O: C>1) compounds are detected by both techniques.

The results suggest that challenges remain in the identification of SOA products using only one technique owing to different capabilities and preferences of instruments. For example, the

electrospray ionisation used in the LC-Orbitrap MS has different ionisation efficiencies and sensitivity to molecules, i.e. more sensitive to carbonyl species in positive mode and carboxylic acid or other polar species in negative mode (Mehra et al., 2020b). The iodide-adduct ToF-CIMS is more sensitive to the polar molecules and sensitivity increases with the addition of a polar functional group in the order of keto-, hydroxy and acid groups (Lee et al., 2014). Thus,


the benefits of using combinatorial analytical techniques enable different insights into components found in SOA particles.

## 3.2 Temporal change in elemental composition

The evolution of gas- and particle-phase oxidation products grouped according to their elemental formulae and carbon numbers is shown in Fig 4. Figure 4a shows that the gas-phase
products are initially dominated by CHON species at the beginning, with a decreasing trend in relative contribution over time, while CHO species are dominant in the particle phase, increasing over time in relative contribution, plateauing after 2.5-h into photooxidation. The $NO_2/NO$ ratio increases to a maximum value after 1 hour and then decreases (Figure S2a), suggesting the CHON species are mostly generated at the beginning with high NO conditions.
In the gas phase, the $C_{8-10}$ ions dominate the CHO and CHON groups. The fraction of $C_{8-10}$ ions in the CHO group increases over time, whereas the fraction of $C_{8-10}$ ions in the CHO group decreases with time. For the particulate oxidation products (Fig 4b), there is a rapid increase to a constant maximum $C_{8-10}$ CHO signal (~42%) and a smaller and slightly reducing $C_{8-10}$ CHON (~15%) over time. There is a significant early increasing signal contribution from compounds
with carbon numbers greater than 15 in the CHO group. The slight increasing fraction of high carbon number compounds ($C \geqslant 16$) is investigated in the particle phase over time. It is not possible to confirm whether compounds with high carbon numbers are generated from either gas-phase chemistry reactions or particle-phase reactions in this study.

## 3.3 Clustering of the oxidation products from FIGAERO-CIMS

The application of HCA to cluster the top 100 CIMS gas-phase and CIMS particle-phase oxidation products (accounting for more than 96% of the total assigned peaks' signals) is shown in Figure 5 and Figure 6, respectively. Since the data are normalized to the highest intensity of each ion, the results are not sensitive to the intensity of raw ion signals. A dendrogram tree is applied to isolate the clusters (Figure S3 for gas phase and Figure S5 for
particle phase). The number of clusters is determined according to whether it already adequately describes the temporal variation and the degree of similarity between clusters. Here, five clusters are chosen for the gas phase and six clusters are chosen for the particle phase according to these criteria. Figure 5a and Figure 6a display the average time series of each cluster calculated by averaging the signals of all ions in every measurement. The time series of
ions in each cluster in the gas and particle phase are shown in Figure S4 and Figure S6





respectively, and formulae information in each cluster is provided in Table S2 and Table S3 respectively. It can be seen that the ions with similar time-series trend are mostly clustered together. The matrix showing the relative Euclidean distance between each cluster pair is provided in Figure 5b and Figure 6b, showing a clear distinction between the clusters, with low

similarity and high confidence in the quality of the clusters.

In the gas-phase HCA, cluster 5 with the highest cluster contribution peaks at the beginning of the experiments (around 1 hour) before reaching a sharp maximum followed by a more gradual decrease until the end of the experiment. The majority of the ions in this cluster are nitrogen-containing. Cluster 4 (containing $C_{10}H_{15}NO_5$ and $C_{10}H_{15}NO_8$) and cluster 5 display similar rates

to their peak values, but different rates of decrease after 1 hour. The temporal profiles of cluster 5 and cluster 4 show similar temporal profiles to the $NO_2/NO$ ratio in Fig S2a. This suggests that these early-generation products are likely formed from the reaction of the OH or $O_3$ initiated peroxy radical with NO rather than with $RO_2$ or $HO_2$ (Eddingsaas et al., 2012b). As shown in Fig 5c and d, cluster 4 and cluster 5 have higher weighted average oxygen numbers

and weighted average carbon number and carbon oxidation state than other clusters.  The high number of oxygen atoms presents indicates these species may have formed via isomerisation of the peroxy radical or secondary OH or $O_3$ chemistry.

Cluster 3 is the second to peak and is dominated by the CHO species, suggesting significant termination by $RO_2$ with $HO_2$ or $RO_2$ at this stage. The lower nC and nO might be caused by

C-C bond scission during secondary photooxidation reactions over time (e.g. $C_9H_{14}O_5$, $C_8H_{12}O_5$, etc), while the higher $\overline{OS_C}$ in cluster 3 is likely caused by the dominant non-nitrogen-containing compounds (or by the fragmentation leading to a lower carbon number whilst retaining oxygen). Cluster 2 is the third cluster to peak and increase over time, consistent with a later generation of products (Eddingsaas et al., 2012a). The weighted average nC, nO and

$\overline{OS_C}$ values exhibit the lowest nC and nO, as shown in Fig 5c and d. Some carbon-hydrogen species (CHO) appear in this cluster, as shown in Table S2, likely produced from the reactions of $RO_2$ with $HO_2$ or $RO_2$. Some nitrogen-containing ions (CHON) with nC≤8 are in this cluster, suggesting they have undergone a higher degree of fragmentation. Cluster 1 shows the slowest formation rate and lowest cluster contribution suggesting they are likely later generation

products.

Particle-phase HCA results are shown in Figure 6, and the formulae in each cluster are listed in Table S3. Cluster 5 exhibits the highest contribution to the total identified peaks signals. It





shows a continuous increase and slight decrease at the end of experiments, suggesting its constituent ions continuously increase in concentration in the particle phase. The majority of

ions in cluster 5 are non-nitrogen-containing with 7 or 10 carbon atoms, and which may partition from the gas phase directly (Zhang et al., 2015). Several dimers (e.g., ion $C_{19}H_{28}O_7$, ion $C_{18}H_{26}O_7$, ion $C_{17}H_{26}O_7$, etc) contribute to cluster 5, which is expected from further accretion reactions (Kristensen et al., 2014; Zhang et al., 2015). The similar temporal profiles between cluster 5 and SOA mass (Figure S2b) indicate cluster 5 contains the most species with

SOA mass. Cluster 4 has the second-highest contribution to the total identified peaks signals. The majority of the ions in this cluster are $C_{9-10}$, suggesting less C-C bond cleavage happened in this cluster. The remaining clusters (cluster 1-3 and cluster 4) show sufficiently different temporal profiles to warrant distinct clusters, but each contains few ions and their cluster contributions to the total signals of identified peaks are lower than 5%.

Many of the top 100 gas-phase oxidation products are also found in the top 100 particle-phase ions. Some ions in the same gas-phase HCA cluster may be distributed into different clusters in the particle phase. For example, ions $C_{10}H_{16}O_3$ and $C_{10}H_{16}O_4$ in the gas-phase cluster 5 are in the particle-phase cluster 2 and cluster 3, respectively. Different clustering between gas- and particle phases indicate that the particle components time-series trend does not unambiguously

relate to the time trend of the corresponding gas-phase ion. Additionally, some ions with high carbon numbers ($nC \geq 16$, e.g., ion $C_{19}H_{28}O_7$, ion $C_{18}H_{26}O_7$, ion $C_{17}H_{26}O_7$, etc) in the particle phase are not included in the top 100 gas-phase ions.

Whilst a common peak-list is used for CIMS data, the HCA is conducted for individual experiments and a single representative experiment is presented here. The effects on clustering

of the variability of concentrations and temporal evolution are not straightforward to understand and present as uncertainty that allows ready interpretation. More than 85 ions are the same in the top 100 for the repeat experiment in both gas and particle phases, accounting for 92% - 99% of the top 100 ions. Gas and particle-phase HCA results for the repeat experiment are provided in Figure S7- S10 and Table S4-S5. The generated cluster name from

the *MATLAB* code differs between experiments (e.g., gas-phase cluster 4 in the representative experiment corresponding to cluster 3 in the repeat experiment), even though their time-series trends and chemical properties are highly similar. In order to easily compare the two experiments, the time-series trends and clusters' fraction in the representative experiment were considered as the benchmark to adjust the cluster name in the repeat experiment for the gas and





particle phase, respectively. The cluster fraction was taken as the particle-phase HCA reference since only six values (peak areas from six particle-phase desorptions) were used to do the particle-phase HCA and the particle-phase temporal profiles will therefore exhibit greater uncertainties than those in the gas phase owing to their lower temporal resolution.

Comparing Figure S7 to Figure 5, it can be seen in the gas phase, cluster 3 in the repeat
experiment contains fewer ions than that in the representative experiment, while cluster 1 in the repeat experiment contains more ions. Figures S7c and d show the different average nC (9.23 vs 8.55) but comparable nO (5.54 vs 5.31) and $\overline{OS_C}$ (-0.12 vs -0.13) in cluster 3 and the different average nC (6.87 vs 7.18) but comparable nO (3.57 vs 3.34) and $\overline{OS_C}$ (-0.03 vs -0.1) in cluster 1 for the repeat and representative experiments, respectively. In the particle phase
(see in Figure S9), the chemical characteristics ($\overline{nC}, \overline{nO}$ and $\overline{OS_C}$) are comparable for clusters with the highest and second-highest contributions between the two experiments, while the time-series profiles for the second-highest contribution clusters are slightly different, decreasing after 1 hour and after 2.5 hours, respectively. The different clustering results for some ions with low normalized signal contributions between the two experiments may indicate that those ions
are likely sensitive to the experimental conditions (e.g., injected α-pinene concentration or seed concentrations), leading to different time-series trends in different experiments. The results suggest that the HCA is a powerful approach to separate ion's time-series behaviour. The chemical characteristics for clusters between the two experiments are comparable, although some individual ions differ in the time-series trend between the two experiments, resulting in
different numbers of compounds in clusters. The differences do not affect the conclusions from the representative experiment due to the low contribution of those ions.

### 3.4 Major contributing ions and their properties

### 3.4.1 Isomeric contributions and evolution in gas and particle phase

The representative oxidation products with relatively high contributions (normalised fractional
signal > 0.5%) from the above HCA particle-phase results were selected to investigate their isomeric contributions, evolution and saturation concentrations in order to elucidate the mechanisms of their potential partitioning to the particle phase during SOA formation from α-pinene photochemistry.

A specific capability of the LC-Orbitrap MS is the ability to separate isomers and quantify their
contributions, not possible with the current CIMS technique. Figure 7a shows all molecular





formulae with a relative contribution more than 1%, demonstrating the separation of isomers with different retention times by LC-Orbitrap MS. Commonly, one isomer dominates the relative contribution at a particular mass, most likely due to the limited formation routes when a single VOC is oxidised. For example, the contribution of formula $C_9H_{14}O_4$ at RT= 5.977 min
is much higher than the isomer at RT= 13.126 min (21.6% vs 0.5%). In the α-pinene system, the dominant contribution of only one isomer in one $m/z$ in offline measurement can support the FIGAERO-CIMS molecular assignment, although the isomers cannot be separated by the FIGAERO-CIMS.

The FIGAERO-CIMS can semi-simultaneously quantify the evolution of the molecular
composition of compounds in the vapor and the condensed phase as shown in Fig 7b and 7c, allowing investigation of gas-to-particle conversion processes. The partitioning behaviour of these compounds was explored by investigating their time series of particle fractions, $F_p$, as shown in Figure 7d and Figure S11. Distinct temporal profiles were observed in each case. For example, $F_p$ increases to maximum stable values for $C_9H_{14}O_4$ after 3.5 hours following
initiation of photochemistry, and for $C_{19}H_{28}O_7$ and $C_{17}H_{26}O_8$ after 1.5 hours, but with $C_8H_{12}O_4$ reducing after a rise in the initial 1.5 hours of photochemistry. As expected, the two compounds in the "dimer" range ($C_{19}H_{28}O_7$ and $C_{17}H_{26}O_8$) have higher particle-phase fractions (almost 1) than those of the "monomer" range oxidation products ($C_8H_{12}O_4$ and $C_9H_{14}O_4$) as shown in Figure 8.

**3.4.2 Saturation concentrations**

The final Fp values from the online measurements were used to calculate the C* as shown in Table 2. The LC-Orbitrap MS chromatography and mass spectrum of compounds are shown in Figure S12. The structural information and estimated $C_i$* based on the LC-Orbitrap MS data are also displayed in Table 2.

The C* calculated from the FIGAERO-CIMS measured $F_p$ was generally within around 2 orders of magnitude of that estimated from the LC-Orbitrap MS-derived molecular structure for most compounds and sometimes much closer. Such agreement could be used to assign classification into the somewhat broad and arbitrarily defined volatility categories (such as semi-volatile, low volatility, intermediate volatility etc) but such utility is questionable.
Moreover, clear discrepancies were noticed for the lowest and highest volatility compounds, as shown in Table 2. It appears that the structural assignment of $C_2H_2O_3$ and $C_3H_2O_3$ using the



MCM provides unreasonable estimates of high volatility, such that such components could not reasonably be expected to be found in the particle phase. The ions detected at this m/z in the FIGAERO-CIMS are likely from desorption fragments of larger ions as their desorption T$max$

are around 83°C and 105°C for $C_2H_2O_3$ and $C_3H_2O_3$, respectively. In any case, the physical meaning of their existence in the particle sample is questionable. It is possible that the accurate determination of the $F_P$ for compounds with large number of carbon atoms ($C_{17}H_{26}O_8$ and $C_{19}H_{28}O_7$) is limited by the detection limit and signal to noise ratio in the gas phase measurements (Lopez-Hilfiker et al., 2016b; Stark et al., 2017). In addition to the difficulties

in measuring partitioning behaviour, challenges remain in its prediction from molecular structure, owing to the acknowledged problems with vapour pressure (and hence C*) estimation, particularly for multifunctional compounds.

Although the C* estimation from the two instruments provides valuable cross confirmation for C* of the targeted compounds, the results suggest that accurate measurement for C* of

compounds remains challenging. The differences in the estimation C* of a compound between the two instruments may be attributable to the different estimation methods. The structural information-based method relies on knowledge of the input of the molecular structure to the vapor pressure estimation model, while the partitioning method is dependent on accurate experimental measurements in the gas-to-particle partitioning process. For the partitioning

approach, the measured particle fraction is likely biased owing to the thermal desorption of larger ions from the filter (Lopez-Hilfiker et al., 2015), or the gas-phase signals of compounds can be over measured owing to the signal-to-noise limitations (Lopez-Hilfiker et al., 2016b; Stark et al., 2017). The current gaps between the two methods require much work to further investigate the C* of compounds in the aerosol particles.

**Table 2.** Molecular information, being ordered in carbon number, and estimated C* of compounds obtained from partitioning theory method and structural information method. The uncertainty of the LogC* from the partitioning theory method was from the uncertainty of measured Fp and was propagated 1 standard deviation.

| Formula | MW | RT (min) | Tentative structure and chemical name | Log C* ($\mu g\ m^{-3}$)[1] | Log $C_{Fp}$* ($\mu g\ m^{-3}$)[2] |
|---|---|---|---|---|---|
| $C_2H_2O_3$ | 74 | n.a |  | 6.76 | 2.48±0.46 |
| $C_3H_2O_3$ | 86 | n.a |  | 8.36 | 2.62±0.13 |





| Formula | m/z | | Structure | | |
|---|---|---|---|---|---|
| $C_4H_8O_4$ | 120 | 0.858 | | 2.48 | 3.92±0.05 |
| $C_5H_8O_3$ | 116 | 2.122 | | 4.68 | 5.51±0.05 |
| $C_5H_8O_5$ | 148 | 0.951 | | 0.96 | 2.38±0.18 |
| $C_7H_{10}O_4$ | 158 | 2.165 | | 5.63 | 3.04±0.65 |
| $C_7H_{12}O_4$ | 160 | 1.623 | | 0.86 | 3.2±0.48 |
| $C_7H_{12}O_5$ | 176 | 1.398 | (Zhang et al., 2018) | 1.15 | 2.08± 0.38 |
| $C_7H_{12}O_5$ | 176 | 2.254 | | 0.46 | |
| $C_8H_{12}O_4$ | 172 | 2.875 | Terpenylic Acid (Claeys et al., 2009) | 3.52 | 2.63±0.42 |
| $C_8H_{12}O_4$ | 172 | 4.536 | Norpinic acid (Winterhalter et al., 2003) | 1.80 | |
| $C_8H_{12}O_5$ | 188 | 1.326 | | 1.26 | 2.09±0.37 |
| $C_8H_{12}O_5$ | 188 | 1.782 | | 4.55 | |
| $C_8H_{14}O_5$ | 190 | 1.94 | | 0.47 | 2.58 ±0.19 |



| Formula | Mass | RT | Structure | Value | Group |
|---|---|---|---|---|---|
| $C_9H_{14}O_4$ | 186 | 5.977 | (Yasmeen et al., 2011) | 1.34 | |
| $C_9H_{14}O_4$ | 186 | 13.126 | carboxylic acid (Yasmeen et al., 2011) | 1.34 | 2.4±0.32 |
| $C_9H_{14}O_5$ | 202 | 2.916 | | 0.49 | |
| $C_9H_{14}O_5$ | 202 | 4.272 | | 1.16 | 2.24±0.75 |
| $C_{10}H_{14}O_4$ | 198 | 2.958 | | 1.58 | 2.4±0.32 |
| $C_{10}H_{14}O_5$ | 214 | 3.432 | | 2.21 | 2.11±0.28 |
| $C_{10}H_{16}O_3$ | 184 | 7.935 | | 3.85 | 3.09±0.45 |
| $C_{10}H_{16}O_4$ | 200 | 4.297 | Hydroxy-pinonic acid (Kristensen et al., 2014) | 3.91 | 2.57±0.41 |
| $C_{10}H_{16}O_5$ | 216 | 5.717 | | 0.93 | |
| $C_{10}H_{16}O_5$ | 216 | 4.948 | | 1.21 | 1.82±0.64 |
| $C_{10}H_{16}O_6$ | 232 | 3.463 | | -0.91 | |
| $C_{10}H_{16}O_6$ | 232 | 6.189 | | 2.07 | 1.73±0.48 |

| | | | | | |
|---|---|---|---|---|---|
| $C_{17}H_{26}O_8$ | 358 | 12.167 | (Kristensen et al., 2014; Demarque et al., 2016) | -2.64 | 1.17±0.38 |
| $C_{19}H_{28}O_7$ | 368 | 14.268 | (Kristensen et al., 2014) | -0.55 | 1.09±0.33 |

Note: [1]Log C* estimated from the molecular structural information-based approach. [2]Log C* estimated from the partitioning theory approach.

## 4. Conclusions

The characterization of SOA chemical composition in the gas and particle phases is an important but complex process, containing potentially thousands of oxidation products. This study demonstrates the capabilities of the combination of FIGAERO-CIMS and LC-Orbitrap MS in the investigation of SOA chemical compositions and their properties in the α-pinene photooxidation reactions.

The distribution of particle-phase products is broadly similar between the FIGAERO-CIMS and LC-Orbitrap MS negative ionisation mode for the α-pinene SOA products. The LC-Orbitrap MS positive ionisation mode provides additional information, with more detected molecular formulae with higher carbon and oxygen numbers. Different insights into components in SOA particles can be gained from the combination of the two techniques.

The gas and particle time-series data from the FIGAERO-CIMS measurement time-series were separated by the hierarchical clustering analysis (HCA), an approach that does not rely on mass calibration. HCA was able to derive cluster properties in terms of average carbon number, oxygen number, oxidation states, for example, enabling interpretation in terms of possible termination pathways of peroxy radicals in the α-pinene photooxidation reactions.

There is substantial uncertainty in the accurate estimation of saturation concentration of compounds. C* estimation from the partitioning method using FIGAERO-CIMS measurements and molecular structure-based method from LC-orbitrap MS enables comparison of the volatility of the targeted compounds. The differences resulting from the two approaches indicates that further work is required to investigate whether the C* determined for



the compounds are consistent with our understanding of the processes determining the time evolution of phase partitioning.

735    There are additional questions that remain to be addressed in future work. First, instrumental sensitivity needs more effort in order to have a better understanding of SOA chemical composition. Second, future work will utilize the analytical approaches from the two instruments to explore SOA oxidation products and their chemical properties from complex systems, such as anthropogenic VOCs or biogenic and anthropogenic mixed VOCs, in the 740    chamber experiments. As this study only assessed a single a-pinene photooxidation system, more work is needed to evaluate how those analytical methods from the two instruments perform with other systems.

## Data availability

The dataset of this study is available on the open dataset of EUROCHAMP programme (https://data.eurochamp.org/data-access/chamber-experiments/).

## 745   Author contributions

GM, MRA, MD, AV, YW and YS conceived the study. MD, AV, YW, and YS conducted the experiments. TJB provided on-site help deploying the FIGAERO-CIMS. KLP and JFH offered help on the LC-orbitrap MS analysis. MD conducted the data analysis and wrote the manuscript with inputs from all co-authors.

## 750   Competing interests

The authors declare that they have no conflict of interest.

## Acknowledgements

The Manchester Aerosol Chamber received funding from the European Union's Horizon 2020 research and innovation programme under grant agreement no. 730997, which supports the 755    EUROCHAMP2020 research programme. The financial support received by MD in the form of a President Doctoral Scholarship award (PDS award) by the University of Manchester is gratefully acknowledged. M.R.A. acknowledges support by UK National Centre for Atmospheric Sciences (NACS) funding. Instrumentational support was funded through the



NERC Atmospheric Measurement and Observational Facility (AMOF). The Orbitrap MS was

760    funded by a capital grant from NERC (CC090). We acknowledge funding for the COMPLEX-

OA project funded by NERC (NE/S010467/1).



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



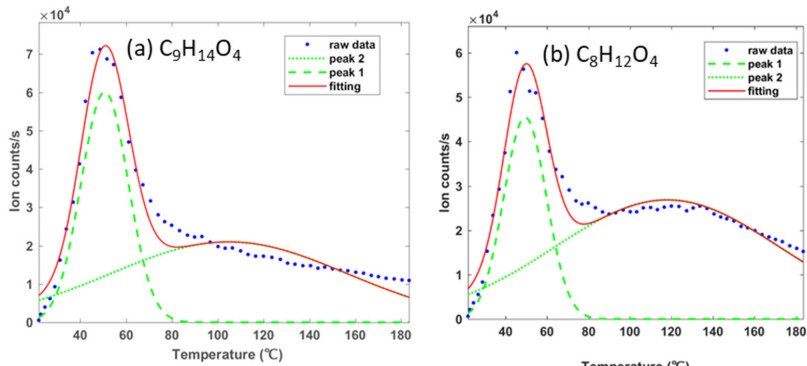

Figure 1. Two examples of an ion with more than a single desorption peak in the FIGAERO-CIMS. Here the peaks at the lower desorption temperature were assigned to monomer $C_9H_{14}O_4$ (a) and $C_8H_{12}O_4$ (b) generated from α-pinene directly and the broader higher temperature peaks being a fragment of another ion at the same mass.

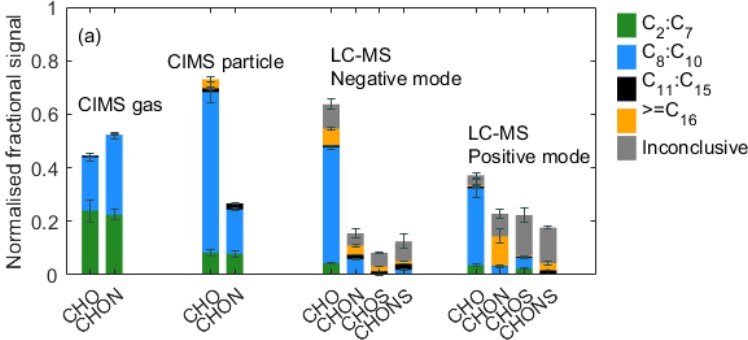





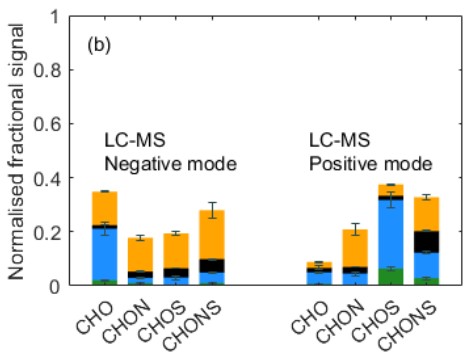

Figure 2. (a) Elemental grouping from CIMS gas phase measurement, CIMS particle-phase measurement, LC-Orbitrap MS in the α-pinene system. The signal of each compound is normalized to all compounds. (b) Elemental grouping for LC-Orbitrap MS inconclusive compounds. The signal of each compound is normalized to the total inconclusive compounds. The colors correspond to the ones in (a). Standard deviations (n=2 for CIMS and n =3 for LC-Orbitrap MS) are indicated by error bars

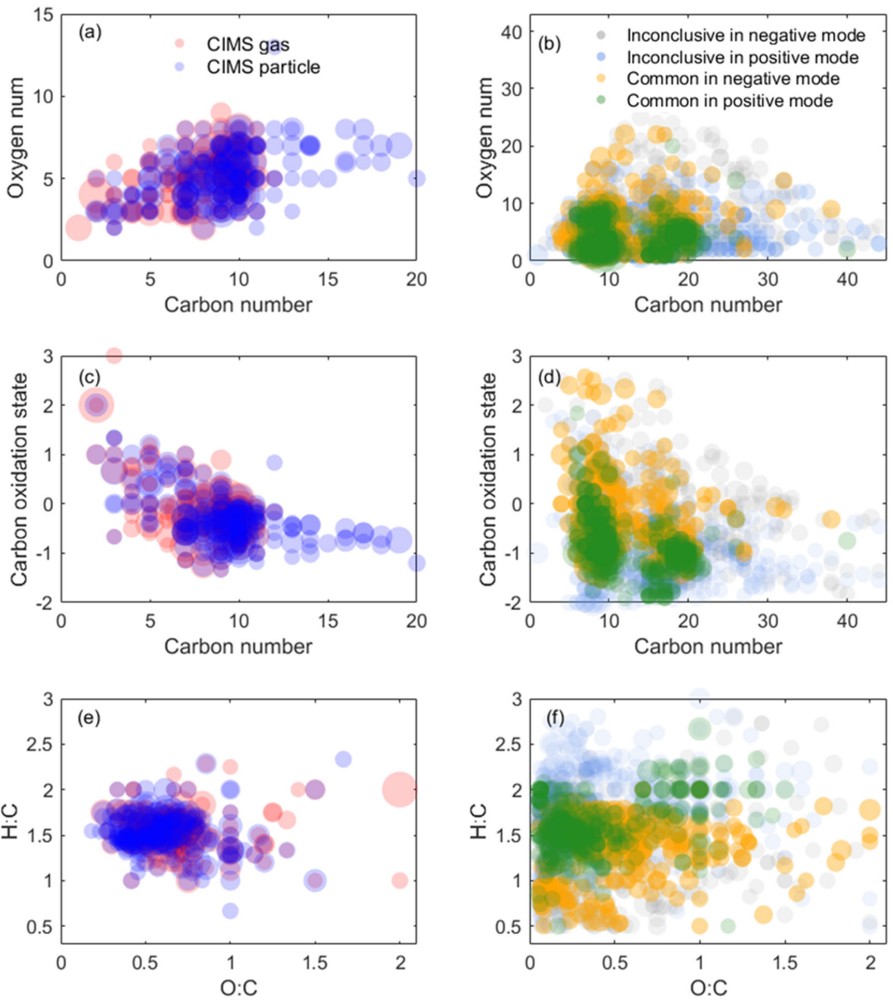

Figure 3. Oxidation products distribution from the representative experiment detected by the CIMS measurement and LC-Orbitrap MS negative and positive mode. (a), (c) and (e) are for the CIMS measurements. (b), (d) and (f) are for LC-orbitrap MS measurement. The symbol size is proportional to the square root of the ion's signal intensity.





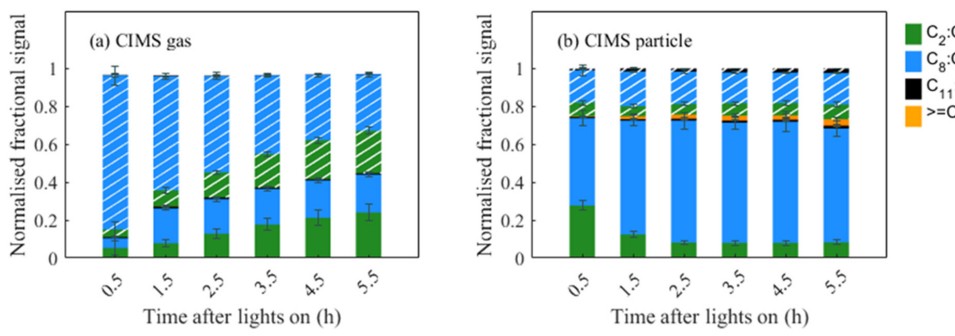

Figure 4. Evolution of chemical compounds in the gas phase and particle phase. Unhatched bars are the CHO group species; hatched bars are the CHON group species. (a) CIMS gas phase; (b) CIMS particle phase. Standard deviations (n =2) are indicated by error bars.

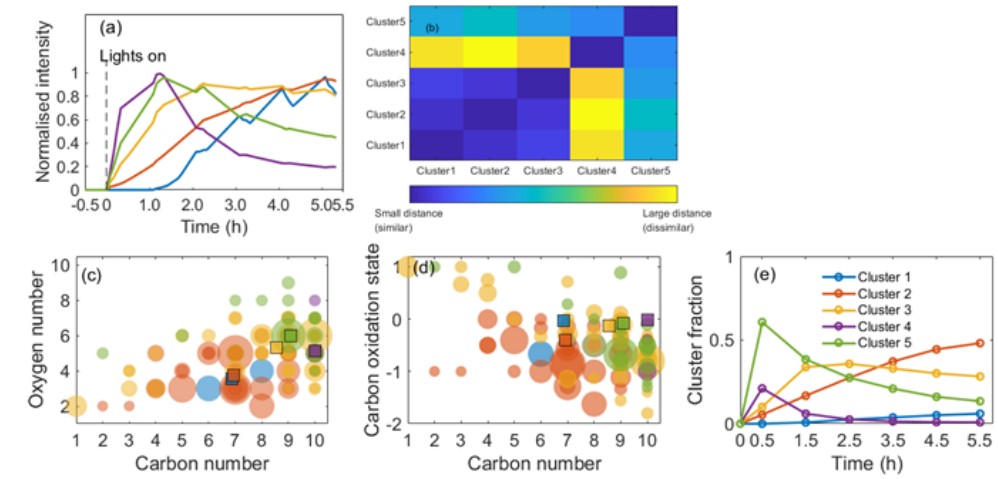

Figure 5. Hierarchical clustering of gas-phase oxidation products from the representative α-pinene system. (a) Time series of each cluster normalized to the highest ions' intensity between 0 and 1. (b) Matrix showing the relative distance between clusters. (c) Carbon number vs oxygen number for each cluster. (d) Carbon number vs oxidation state for each cluster. (e) Time series of the sum of ions' normalized signal

fractions to the total signal in each cluster. Note that the square symbols represent the contribution weighted average carbon numbers, oxygen numbers or and $\overline{OS_C}$ in each cluster. The colors correspond to the ones in (a).



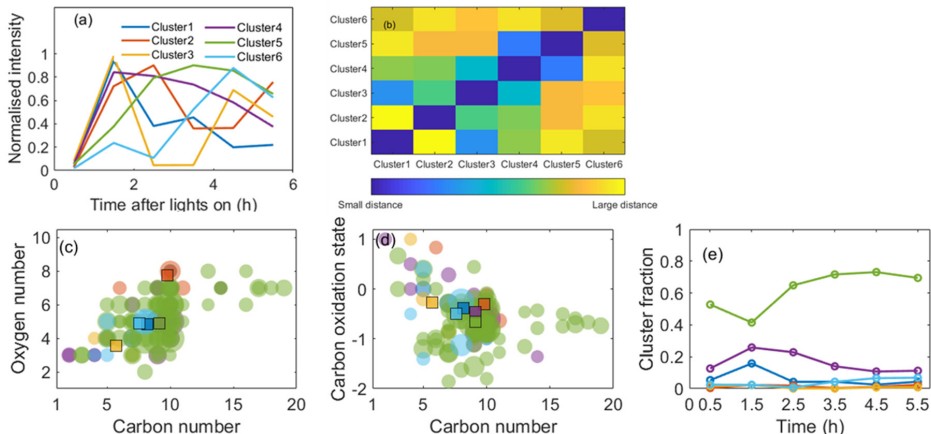

Figure 6. Hierarchical clustering of particle-phase oxidation products for the representative α-pinene system. (a) Average time series of each cluster normalized to the highest ions' intensity between 0 and 1; (b) Matrix showing the relative distance between clusters. (c) Carbon number vs oxygen number for each cluster. (d) Carbon number vs oxidation state for each cluster. (e) Time series of the sum of ions' normalized signal fractions to the total signal in each cluster. Note that the square symbols represent the contribution weighted average carbon numbers, oxygen numbers or and $\overline{OS_C}$ in each cluster. The colors correspond to the ones in (a).

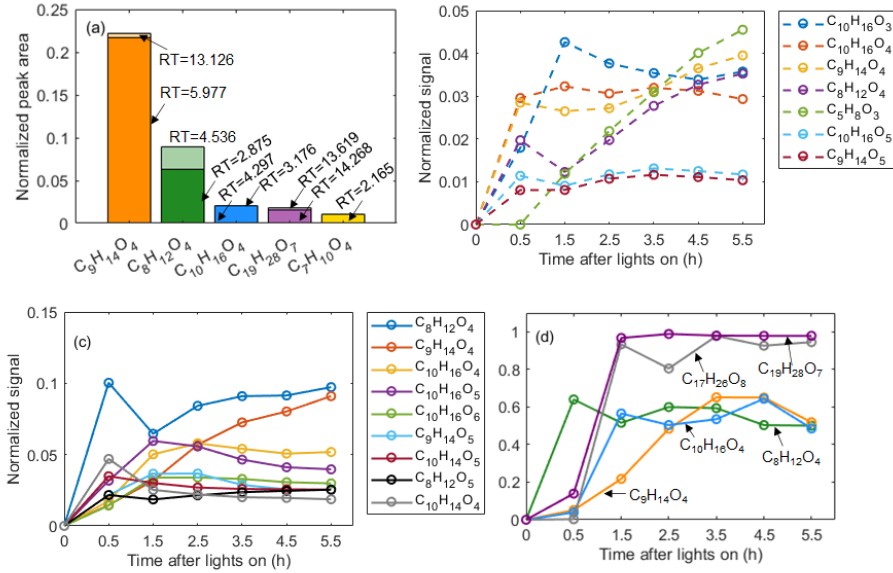

Figure 7. (a) Isomeric fractions of ions from LC-Orbitrap MS analysis; (b) Time series of compounds in the gas phase from CIMS gas-phase measurements. (c) Time series of compounds in the particle phase from CIMS particle-phase measurements. (d) Temporal profiles of particle fractions for a few identified organic molecules generated from α-pinene photooxidation reactions.