# Peer review of "Combined application of Online FIGAERO-CIMS and Offline LC-Orbitrap MS to Characterize the Chemical Composition of SOA in Smog Chamber Studies"

_Atmospheric Measurement Techniques, 2021_

## Author Comment (AC1)

Atmos. Meas. Tech. Discuss., referee comment RC2
https://doi.org/10.5194/amt-2021-420-RC2, 2022

[Figure]

**Comment on amt-2021-420**

Anonymous Referee #1
* * *
Referee comment on "Combined application of Online FIGAERO-CIMS and Offline LC-Orbitrap MS to Characterize the Chemical Composition of SOA in Smog Chamber Studies" by Mao Du et al., Atmos. Meas. Tech. Discuss.,

https://doi.org/10.5194/amt-2021-420-RC1, 2022
* * *
Du et al. combined online FIGEROI-CIMS with offline LC-orbitrap MS to measure products from a-pinene oxidation. They compared the chemical composition measured by the two instruments, performed hierarchical clustering based on temporal behaviors, and compared volatility using CIMS partitioning measurements and structural information derived from LC-orbitrap MS. The combination of the two techniques provides insights into structures, isomer information, and uncertainties in vapor pressure estimations. The manuscript is generally well written with clear presentation of the results. I recommend publishing the manuscript after a minor revision.

> We appreciate the reviewer for your time and effort in providing comments for our manuscript. we addressed each comment point by point. For clarity, we kept the reviewer's comment in black. Our answers are marked in deep blue and changes to the manuscript are in deep blue and Italic. Line numbers in the responses refer to the revised manuscript.

My comments are listed below:

- It is unclear from the manuscript what OH source the authors used.

  > The initial OH source was generated through the photolysis of the injected $NO_2$. During the experiment, photolysis of ozone and unsaturated VOC ozonolysis (or potentially via OH recycling) can also generate OH source. The irradiance for photolysis of both $NO_2$ and $O_3$ is provided by two 6 kW Xenon arc lamps and 5 rows of halogen lamps as described in Shao et al. (2022). This description has been added to the manuscript in L182th-186th.

- Did the authors report both iodide clustered species and de-clustered species? In Table S2-S4, there are some open-shell products. This makes the reviewer wonder if those open-shell products were formed in the chamber or just de-clustered species in the instrument.

  > We reported that all species in the Table S2-S4 are iodide clustered species. In order to make it clearer. The sentence *'All ions are reported as iodide clustered species.'* has been added in the caption of tables S2-S4.

There are a few open-shell products such as $C_{10}H_{15}O_5$ and $C_9H_{15}O_5$ which were reported by previous literature (Shilling et al, 2009). Tthose products are generated from the oxidation of a-pinene. However, their detection in our study is likely from either the formation via the chamber, clustering with $I^-$ and being detected, or from being de-clustered species in the instrument. More work is needed to distinguish their specific source in future.

- Did the authors observe any delay of signals due to losses to tubing or IMR surface (especially for sticky compounds) in the onset period? Would the delay affect the shape of the time series and thus affect the hierarchical clustering?

  There were delayed signals for some detected compounds in the gas and particle phases, respectively. In this study, we addressed this issue by subtracting the 'instrumental background' for the two phases. Also, the data processing method was validated by Voliotis et al. (2021). Although background correction for all ions was the same, the possible difference in the delayed signals for different compounds may affect the time evolution and thus affect the clustering to different degrees. We acknowledge the reviewer pointed this out and future work is needed to determine the magnitude of the effect. . The detail about the background correction in gas and particle phases was shown below:

  For the gas phase, we employed the fast background measurement (it was stated in the L220 'The instrument was flushed with ultra-high purity nitrogen (UHP, 99.999% purity, $N_2$) for 0.2 min every 2 min during each gas-phase measurement, which acted as the gas-phase instrumental background.' The pure signal of detected compounds was obtained by the signal of 2 min measurement subtracting the signal from the 0.2 min instrumental background to remove the influence of delayed signals in the gas phase.

  In the particle phase, there were delayed signals from the last gas-phase measurement at the beginning of the thermogram period. Thus, the signal in the first 60-90s regarded as the instrumental background was subtracted from the ion signals. As stated in the L246th-250th of the manuscript, 'For the particle phase, the signal in the first 60 - 90s with relatively low and stable signals was considered as the instrumental background, enabling interference between the gas and particle mode switching to be removed (Voliotis et al. 2021).'

- What were the differences in mass loadings of materials collected on the FIGAERO filter vs on the LC filter due to the difference in collection time? Could any of the differences in composition (e.g., high carbon number compounds) be explained by a lower loading that is close to the detection limit in the CIMS?

  The mass loadings of materials on the filters for FIGAERO-CIMS and LC-MS analysis were indeed different. The mass loading of the filter for FIGAERO-CIMS was around 1.55μg ((1slpm *30 min * $10^{-3}$/12 $m^3$ *620 μg; Note: 1slpm is the sampling flowrate; 30 min is the sampling time; 12 $m^3$ is the estimated final chamber volume; 620μg is the average mass of chamber from the SMPS measurement). For the LC-MS analysis, one-quarter of the filters collected at the end of experiments were used with the mass loading of ~166 μg (AS seeds + aerosols) from the SMPS measurement.

  There are a number of factors that could result in the differences in the composition, including thermal desorption of SOA in the filters or the selectivity and sensitivity of

instrumental ionisation methods toward compounds (Stark et al., 2017; Mehra et al., 2020; Voliotis et al., 2021) and possibly differences in the limits of detection of the instruments as suggested by the reviewer. It is not possible to attribute the differences in composition to a specific cause in our experiments, but this should be the focus of future work.

- The reviewer suggests that the authors include a figure to show the time series of the precursor in the experiment. It will be useful to know the decay of the precursor along with the formation of products to understand first-gen vs later-gen products.
  The figure has been added to the supplementary information in Figure S3.

**References:**

Shilling, J. E.; Chen, Q.; King, S. M.; Rosenoern, T.; Kroll, J. H.; Worsnop, D. R.; DeCarlo, P. F.; Aiken, A. C.; Sueper, D.; Jimenez, J. L.; Martin, S. T. Loading-dependent elemental composition of α-pinene SOA particles. *Atmospheric Chemistry and Physics* **2009**, *9*, 771-782.

Mehra, A., Wang, Y., Krechmer, J. E., Lambe, A., Majluf, F., Morris, M. A., Priestley, M., Bannan, T. J., Bryant, D. J., Pereira, K. L., Hamilton, J. F., Rickard, A. R., Newland, M. J., Stark, H., Croteau, P., Jayne, J. T., Worsnop, D. R., Canagaratna, M. R., Wang, L., and Coe, H.: Evaluation of the chemical composition of gas- and particle-phase products of aromatic oxidation, Atmospheric Chemistry and Physics, 20, 9783-9803, 10.5194/acp-20-9783-2020, 2020.

Shao, Y., Wang, Y., Du, M., Voliotis, A., Alfarra, M. R., Turner, S. F., and McFiggans, G.: Characterisation of the Manchester Aerosol Chamber facility, Atomospheric measurement techniques, https://doi.org/10.5194/amt-15-539-2022, 2022.

Stark, H., Yatavelli, R. L. N., Thompson, S. L., Kang, H., Krechmer, J. E., Kimmel, J. R., Palm, B. B., Hu, W., Hayes, P. L., Day, D. A., Campuzano-Jost, P., Canagaratna, M. R., Jayne, J. T., Worsnop, D. R., and Jimenez, J. L.: Impact of Thermal Decomposition on Thermal Desorption Instruments: Advantage of Thermogram Analysis for Quantifying Volatility Distributions of Organic Species, Environ Sci Technol, 51, 8491-8500, 10.1021/acs.est.7b00160, 2017.

Voliotis, A., Wang, Y., Shao, Y., Du, M., Bannan, T. J., Percival, C. J., Pandis, S. N., Alfarra, M. R., and McFiggans, G.: Exploring the composition and volatility of secondary organic aerosols in mixed anthropogenic and biogenic precursor systems, Atmospheric Chemistry and Physics, 21, 14251-14273, 10.5194/acp-21-14251-2021, 2021.

---

## Author Comment (AC2)

(Shilling et al., 2009)

[Figure]

Atmos. Meas. Tech. Discuss., referee comment RC2
https://doi.org/10.5194/amt-2021-420-RC2, 2022

[Figure]

**Comment on amt-2021-420**

Anonymous Referee #2
* * *
Referee comment on "Combined application of Online FIGAERO-CIMS and Offline LC-Orbitrap MS to Characterize the Chemical Composition of SOA in Smog Chamber Studies" by Mao Du et al., Atmos. Meas. Tech. Discuss.,

https://doi.org/10.5194/amt-2021-420-RC2, 2022
* * *
The manuscript titled "Combined application of Online FIGAERO-CIMS and Offline LC-Orbitrap MS to Characterize the Chemical Composition of SOA in Smog Chamber Studies" by Du and Co-authors characterizes the chemical composition of SOA using a combination of two state-of-the-art mass spectrometry techniques: FIGAERO-CIMS (online), and LC- Orbitrap (offline). The chemical system analyzed is the photooxidation of α-pinene using an atmospheric chamber. The authors use hierarchical clustering of the time series of gas-phase and particle-phase oxidation products to get an insight into the phase partitioning of individual molecular compositions and to inform the targeted analysis for the LC orbitrap.

The distribution of particle-phase products is found broadly similar between the FIGAERO-CIMS and LC-Orbitrap MS negative ionization mode.

The hierarchical clustering analysis allowed the identification of cluster properties in terms of average carbon number, oxygen number, oxidation states, for example, enabling interpretation in terms of possible termination pathways of peroxy radicals in the α-pinene photooxidation reactions.

Saturation concentrations of compounds ($C^*$) were carried out using the FIGAERO CIMS and LC orbitrap independently. The substantial differences resulting from the two approaches indicate that further work is required to investigate $C^*$.

The article is well written and presents an interesting and substantial amount of work that is very relevant, of high interest for the scientific community, and well in line with the journal scope.

I recommend publication after the following minor revisions.

We thank the reviewer for your time and effort to provide comments/suggestions on our work. Your contribution is much appreciated.

Please find the following replies to each one of the comments/suggestions. For clarity, we kept the reviewer's comment in black. Our answers were marked in deep blue and changes to the manuscript were in deep blue and Italic. Line numbers in the responses refer to the

revised manuscript.

Line 161: "The … 80C". "relativity" should be "relative" and it should be in the past tense, not future.

It has been changed into 'relative'. The tense has been changed to the past. The sentence is shown here for your convenience.

'*The relative humidity in the chamber was adjusted by the custom-built humidifier which comprises a 50L tank fed with ultra-pure water (resistivity ≥18.2 MΩ-cm), generating water vapour using an immersion heater that heats the water to ~80°C.*'

Line 183: "the filters … contaminants". It would be good to add a sentence about how (or

if?) this process was quantified as well some more information about the loading of the filters.

The following sentence was added in the L194-196th in the manuscript. '*The chemical composition of filters was characterized and expressed as peak area through the offline LC-MS analysis. The ratio of largest residual contaminant that was removed from the sample by the blank subtraction is 188, indicating the much lower loading of blank filters after the pre-treatment.*'

Line 202: "characterized for 30 mins" this is unclear. I recommend changing/expanding the sentence to better explain what that entails.

we rephrased this sentence. It reads as from Line 209th '*Gas-phase species are sampled via a 0.5 m ¼" I.D. PFA tubing at 1 standard litre per minute (SLM) from the chamber. The gas sampling was performed for 30 mins.*'

Line 210: "The gas-phase data … chamber background". This is unclear. Please change/extend the sentence to explain what the authors mean. The first measurements were used to correct gas-phase data for the background?

Yes, the first measurements were used to correct gas-phase measurement data. In L213th, it was stated that '*the first cycle in the cleaned chamber condition*'. And therefore, the data collected during this period were considered as the chamber background. In order to make it clearer, we adjusted the sequence of the description of gas-phase background correction. It read as:

'*Six gas phase and particle phase cycles were carried out during each experiment. The sample blanks were collected by two additional gas and particle-phase cycles before the initiation of photooxidation. The first cycle was conducted in the clean chamber condition before injection of reactants. This was followed by the second cycle after all species (VOC, seeds and NOx) had been injected. The gas-phase data collected in the first cycle was employed for the gas-phase "chamber background". The instrument was flushed with ultra-high purity nitrogen (UHP, 99.999% purity, N₂) for 0.2 min every 2 min during each gas-phase measurement, which acted as the gas-phase "instrument background". To remove the influence of the seed particles, the particle-phase data collected in the second cycle were used as the particle-phase background correction.*'

Line 453: "5.5 hours into the photochemistry experiment." Please add why 5.5 hours was chosen.

In each experiment, the chamber was irradiated for 6 hours after injection of reactants. It was stated in L210th in section 2.2.1 that there were six gas and particle phases over the experiments (0.5h, 1.5h, 2.5h, 3.5h, 4.5h, 5.5h) for online FIGAERO-CIMS measurements, as shown in Figure 1. It can be seen that the 5.5 hour point was the end of the last cycle of gas-phase measurement and commencement of the final cycle of particle-phase

measurements for the online FIGAERO-CIMS measurements. As stated in the manuscript, particles were collected onto the filter in the previous gas-phase measurements, thus the particles analysed started from 5.5 hour and were collected from 5.0 to 5.5 hours. Meanwhile, the filters employed for offline LC-MS analysis were collected from the chamber after 6-hour reactions. In our study, the 5.5 hours is the final cycle for the FIGAERO-CIMS measurements, so it was chosen to compare with the results from the offline analysis. We changed the description as below:

'*Offline filters characterized by LC-Orbitrap MS are shown in Fig 2 from negative and positive ionization mode alongside the last cycle of gas and particle-phase measurements (5.5 hours after lights on ) by the CIMS*'

[Figure]

Figure 1. Gas and particle cycles of FIGAERO-CIMS over the 6 hour experiments.

Line 529-532: "Figure 4a. photoxidation". The sentence is unclear. "increasing over time" seems to be referred to the gas phase. Please rephrase to make sure the reader does not get confused.
It has been changed into: '*Figure 4a shows that the gas-phase products are initially dominated by CHON species at the beginning, with a decreasing trend in relative contribution over time. In Figure 4b, CHO species are shown to dominte in the particle phase, increasing over time in relative contribution, plateauing after 2.5-h into the experiment.*'

Figure S3: the x-axis labels are not legible because are too crowded. Please modify the graph to make them legible (have only a subset?) or remove them if they are not necessary to understand the graph.
We think it would be better to remove it, as the molecules information of each cluster was included in Table S2-S4.
The figures' sequence in the manuscript and SI was also adjusted.

Line 734: "instrumental sensitivity needs more effort" this sentence is unclear. Do the authors mean that the sensitivity needs to improve?
We thank the reviewer for pointing this out. We agree that this sentence was not clear. The last paragraph has been changed into '*The selectivity and instrumental sensitivity make it impossible to use one instrument to capture all oxidation products, thus it strongly recommends the combination of instruments in order to better explore the evolution of SOA chemical composition. Additionally, future work will report SOA oxidation products and their chemical properties from more complex systems utilizing the analytical approaches from the two instruments, such as the chamber photo-oxidation of anthropogenic VOCs or biogenic and anthropogenic mixtures of VOCs (Voliotis et al., 2022). As this study only assessed a single a-pinene photooxidation system, more work is needed to evaluate how*

*those analytical methods from the two instruments perform with other systems.'*

Figure 5: I recommend adding "Gas-phase" somewhere on the plot.
  The 'Gas phase' has been added in Figure 5.
Figure 6: I recommend adding "Particle phase" and "FIGAERO CIMS" somewhere on the plot.
  The 'Particle-phase' has been added in Figure 6.

**Reference:**

Shilling, J. E., Chen, Q., King, S. M., Rosenoern, T., Kroll, J. H., Worsnop, D. R., DeCarlo, P. F., Aiken, A. C., Sueper, D., Jimenez, J. L., and Martin, S. T.: Loading-dependent elemental composition of α-pinene SOA particles, Atmospheric Chemistry and Physics, 9, 771-782, 10.5194/acp-9-771-2009, 2009.

Voliotis, A., Du, M., Wang, Y., Shao, Y., Alfarra, M. R., Bannan, T. J., Hu, D., Pereira, K. L., Hamilton, J. F., Hallquist, M., Mentel, T. F., and McFiggans, G.: Chamber investigation of the formation and transformation of secondary organic aerosol in mixtures of biogenic and anthropogenic volatile organic compounds, Atmospheric Chemistry and Physics Discussions, 10.5194/acp-2021-1080, 2022.

---

## Author Response (AR2)

Thank you for your attention to the reviewer comments. I have a few minor items that still require attention before publication.

We appreciate the editor's time and efforts. We revised the comments one by one as showing follow.

Please indicate units of VOC:NOx. I assume it is ppb:ppb, and not ppb-C:ppb, but please clarify.

Yes, the unit is ppb/ppb, which has been added in the manuscript.

I don't understand this sentence at lines 196-198; please clarify: "The ratio of the largest residual contaminant that was removed from the sample by the blank subtraction is 188, indicating the much lower loading of the blank filters after the pre-treatment"

It means compounds such as compound A in the blank will be removed if the compound meet the three criteria (the same formula, the RT difference is less than 0.1mins and the same molecular weight) and then it was removed from samples. The sentence was changed into *'The ratio of peak areas for the contaminant in the samples to that in the blank filter is 188, indicating much lower loading of the blank filters after the pre-treatment.'*

Line 230 - this sentence is repeated twice: "To remove the influence of the seed particles, the particle-phase data collected in the second cycle were used as the particle-phase background correction"

thanks, delete the repeat one sentence in the manuscript.

Line 556 - "dominate" is misspelled

It was changed.

Fig 5 and 6 - the label for panel (b) is very small and could be large
It was changed into the clear captions.
The Font size of labels and caption were changed into lager size.